# Symmetry-controlled edge states in the type-II phase of Dirac photonic lattices

Georgios G. Pyrialakos [1], Nora Schmitt[2], Nicholas S. Nye[1,3], Matthias Heinrich [2], Nikolaos V. Kantartzis [1], Alexander Szameit [2] & Demetrios N. Christodoulides [3✉]

The exceptional properties exhibited by two-dimensional materials, such as graphene, are rooted in the underlying physics of the relativistic Dirac equation that describes the low energy excitations of such molecular systems. In this study, we explore a periodic lattice that provides access to the full solution spectrum of the extended Dirac Hamiltonian. Employing its photonic implementation of evanescently coupled waveguides, we indicate its ability to independently perturb the symmetries of the discrete model (breaking, also, the barrier towards the type-II phase) and arbitrarily define the location, anisotropy, and tilt of Dirac cones in the bulk. This unique aspect of topological control gives rise to highly versatile edge states, including an unusual class that emerges from the type-II degeneracies residing in the complex space of k. By probing these states, we investigate the topological nature of tilt and shed light on novel transport dynamics supported by Dirac configurations in two dimensions.

[1] Department of Electrical and Computer Engineering, Aristotle University of Thessaloniki, GR-54124 Thessaloniki, Greece. [2] Institute of Physics, University of Rostock, Albert-Einstein-Str. 23, 18059 Rostock, Germany. [3] College of Optics & Photonics-CREOL, University of Central Florida, Orlando, FL 2816, USA. ✉email: demetri@creol.ucf.edu

The experimental discovery of graphene ushered in a new era in the development of two-dimensional (2D) materials, which were only previously theorized[1–3]. In graphene-based systems, the bulk excitation spectra exhibit a linear dispersion relation—a Dirac cone—akin to that governing massless Dirac fermions in quantum relativistic theories. These conical band degeneracies act either as a source or a sink of Berry curvature in momentum space and under broken time-reversal symmetry their pairs favor the emergence of a well-defined Chern number in the first Brillouin zone. Such inherent topological properties are known to lead to a wealth of unusual phenomena like the quantum Hall effect, anomalous magnetoresistance, and the appearance of edge states in terminated systems[4–8], that is of crucial importance to current transport dynamics[9,10].

The observation of type-I massless Dirac and Weyl fermions in solid state systems[11–15] has spurred an intense research activity in other fields like, for example, photonics, acoustics, and other bosonic settings[16–22]. In this vein, generalized forms of the Weyl Hamiltonian that either modify or shift the underlying symmetries of the fermionic model have attracted considerable attention. In certain classes of systems displaying a broken Lorentz invariance, the Fermi surfaces become conical (type-II) instead of point-like as in type-I, owing to the underlying strong tilt of the associated Weyl cones. Such type-II Weyl fermion excitations have been lately observed in semimetals, e.g., $WTe_2$, $MoTe_2$, $PdTe_2$, along with Fermi arcs and unusual chiral photogalvanic effects[23–30]. An issue of interest is how the rich topological properties of type-II Weyl and Dirac systems can alter the properties and dynamics of localized edge states. In this respect, to what extent will this class of edge states occupy the Brillouin zone (with regard to the type-II Dirac cones), and what parameters will determine their mobility? If so, can such interactions be observed in a purely bosonic environment[31–34]?

In this work, we explore the type-II Dirac degeneracies emerging in a 2D all-dielectric, non-centrosymmetric, photonic topology. To this aim, adjustable waveguide chains were introduced between the main lattice sites of a centered-square lattice in order to reproduce the molecular bond structure, characteristic of the class of artificial carbon allotropes known as graphynes[35–37]. These chains enable a controlled variation of the effective lattice intercouplings and provide access to the full parameter space of the quasi-relativistic model. By studying a number of archetypical examples, we observe the unique and rich edge state dynamics that arise from the type-II cones and demonstrate how these band degeneracies can relocate outside the first Brillouin zone and generate tilted edge states that extend throughout all k space. Fundamentally, the exclusive use of single-mode elements in the optical lattice allows for a full characterization of the bulk and edge dynamics through a single-orbital, tight-binding formalism. This produces a simple yet highly versatile and universal archetype for the exploration of the subtle physics of quantum relativistic effects in a variety of optical and molecular systems[38–40].

## Results

**Dirac Hamiltonian in a chained lattice model.** The relativistic Dirac equation provides a fundamental description of electron dynamics in many diatomic systems, in which the Dirac spinor is associated with the eigenmode vector of the tight-binding, two-band representation. A class of specially synthesized multi-element arrangements may also promote an analogous behavior, while introducing new degrees of freedom to the relativistic model (e.g., graphynes[35,41]). In these arrangements, the effective $2 \times 2$ Dirac Hamiltonian can be obtained by identifying two principal elements in the unit cell while extracting a reduced coupling representation. Employing a similar approach, we examine the tight-binding model of a centered square lattice (Fig. 1) under a broad generalization of its discrete representation, which, as shown in the Supplementary Note 1, can produce the extended Dirac Hamiltonian,

$$H(\mathbf{k}) = u^{\mathrm{T}} \cdot (\mathbf{k} - \mathbf{k}_{\mathrm{D}})I + \left[ u^{\mathrm{D}}(\mathbf{k} - \mathbf{k}_{\mathrm{D}}) \right] \cdot \boldsymbol{\sigma} + m'\sigma_z, \quad (1)$$

around finite regions at the $(\mathbf{k}_{\mathrm{D}}, -\mathbf{k}_{\mathrm{D}})$ points. In this tight-binding description $\mathbf{k}$ is the wave vector, $\boldsymbol{\sigma} = (\sigma_x, \sigma_y)$ and $\sigma_z$ represent Pauli matrices, and $I$ stands for the identity matrix. All key variables in Eq. (1) are associated with individual symmetries, whose status (broken/unbroken) is strictly linked to the five lattice parameters ($t_1 - t_4$, $t_s$). In particular, the mass term $m'$ is non-zero if the effective P-(sublattice) symmetry or T-symmetry is disturbed, resulting in a gap opening at the Dirac degeneracy. Here, the $2 \times 2$ matrix $u^{\mathrm{D}}$ term determines the multiplication constants of ($k_x\sigma_x$, $k_y\sigma_x$, $k_x\sigma_y$, $k_y\sigma_y$) and hence the anisotropy (orientation/ellipticity) of the Dirac cone. The elements of $u^{\mathrm{D}}$, along with the cone position $\mathbf{k}_{\mathrm{D}}$, are regulated via an explicit manipulation of the hopping terms $t_1 - t_4$ (see Supplementary Eqs. (10)–(18) in Supplementary Note 1) while, in turn, the $t_s$ hopping term independently governs the $u^{\mathrm{T}}$ vector, thus

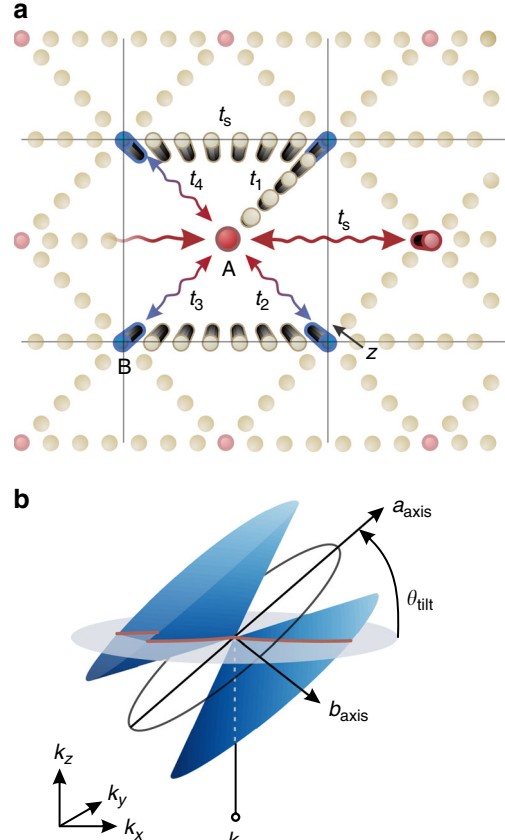

**Fig. 1 Chained square lattice of photonic waveguides and tilted type-II Dirac cone. a** The anisotropic square lattice with five independent hopping terms between sites A and B. Each term is associated with a unique waveguide chain that regulates its magnitude (here, we explicitly draw $t_1$ and $t_s$). Light propagates towards the z direction, with momentum $k_z$ imposed by the eigenvalues of the quasi-relativistic model. **b** The generalized Dirac cone, located at $\mathbf{k}_{\mathrm{D}}$, is characterized by the amount of anisotropy ($a/b$) and degree of tilting ($\theta_{\mathrm{tilt}}$). The plane of anisotropy is in general separate from the $\theta_{\mathrm{tilt}}$ plane. These properties are directly governed by the hopping terms introduced in **a**.

promoting a transition to the type-II phase by disturbing the Lorentz invariance of Eq. (1).

In a diatomic square lattice (i.e., a lattice with direct coupling between sites A and B) the diagonal hopping terms ($t_1 - t_4$) have equal magnitude and phase defined by the excitation wavenumber and distance between the central nodes. To extend this characterization we introduce secondary sites between the central nodes, as shown in Fig. 1, forming a chained lattice. Utilizing the same fundamental tight-binding formalism (see Supplementary Note 1), these chains exhibit a unique ability to control the hoping terms of Fig. 1, breaking the fourfold symmetry of the square topology. Most importantly, they enable arbitrary perturbations to Eq. (1) under unbroken sublattice symmetry (i.e., maintaining a gapless state) while preserving the geometrical features of the unit cell. In the absence of nontrivial topological transitions, a gap at the Dirac cones is, herein, avoided through the symmetric placement of the nodes around each chain's center, thereby nullifying the effective mass. Furthermore, these same bonds allow for interactions beyond nearest-neighbors, which is impossible to accomplish in conventional two-atom models, giving rise to the Lorentz-violating term of Eq. (1). These attributes provide complete access to the parameter space of the Dirac Hamiltonian in terms of freely moving ($\mathbf{k}_D$), rotating ($u^D$), and tilting ($u^T$) the Dirac cones in the bulk, without being exclusively restrained on the high-symmetry lines. Therefore, the chained lattice is not only able to define a group isomorphism with every general perturbation of the honeycomb unit cell (e.g., strained graphene) but also extend this characterization even further.

**The edge states of ribbon-like topologies.** A consistent way to put the aforementioned claims to the test is to study edge states that are topologically connected to the Dirac representation; a field of substantial theoretical and applicative interest in its own right. The topological characteristics of the bands in the bulk are revealed by determining the Zak phase[42], $\gamma$, enclosing the Dirac cones at the $\mathbf{k}_D$ points. This is explicitly defined by the phase of the anti-diagonal terms of the effective Hamiltonian, $H_{12} = |H_{12}| e^{\phi(\mathbf{k})}$, through $\gamma(C) = i \oint_C \nabla_{\mathbf{k}} \varphi(\mathbf{k}) \cdot d\mathbf{k}$, where $C$ is a path in the $\mathbf{k}$ space. The existence of edge states is related to the appearance of discontinuities in $\phi(\mathbf{k})$, which emerge from and end at the $\mathbf{k}_D$ points of a Dirac cone pair (see Supplementary Note 2). The existence and extent of these regions are dictated by the $u^D$ term of the effective Hamiltonian and can be controlled via the $t_1 - t_4$ hopping terms of the lattice model (see Supplementary Fig. 2). In contrast, the type-II ($u^T$) term and its effective counterpart in the lattice model (the $t_s$ hopping term) do not disturb this topological characterization. Therefore, it follows that a type-II terminated lattice, with an arbitrarily enhanced magnitude for $u^T$, should support its own class of edge states.

To study these states, we design a photonic analog of coupled waveguides, a system where light propagates according to the dynamic laws of Schrödinger's equation (see also Supplementary Note 4). In this framework, we consider a simple lattice formed by retaining only three main bonds in its unit cell (namely, $t_4 = 0$ and $t_s = 0$ in the entire $\mathbf{k}$ space), as shown in the unit cell of Fig. 2a. The existing bonds ($t_{1-3}$) are regulated by the $\Delta n_{1-3}$ variables, defined as the refractive index difference of the central waveguide pairs. In a uniform system ($\Delta n_{1-3} = \Delta n_0$, $t_{1-3} = t_0$), a pair of slightly anisotropic Dirac cones of the type-I phase appears at $\mathbf{k}_D = (\pm 2\pi/3, \pm 2\pi/3)$, as depicted in Fig. 2b. To produce a bearded set of edge states, arising from the discontinuous line of Fig. 2c, (Zak phase), one has to accordingly terminate the chained lattice. The unit cell comprises a total of 14 elements, indicating a variety of possible terminations. Here, we

terminate the lattice as shown in Fig. 2a (see also Supplementary Note 3), whose band diagram is depicted in Fig. 2d. Of the four parameters affecting the Dirac cones in the bulk (location, tilt, orientation, and anisotropy), only the first two are associated with inherent properties of the edge stats.

In order to demonstrate control over the location of the Dirac degeneracies, we modify the refractive index difference of the middle waveguide pairs in chain 3 ($\Delta n_3$). This corresponds to a proportionate increase of the potential term (in the Schrödinger definition) and as a result, stronger bonding between the four elements in the chain (in the tight-binding representation). In the effective model, the magnitude of the $t_3$ hoping term increases, disturbing further the rotational symmetry of Eq. (1), which causes the Dirac points to relocate in momentum space. Beyond a critical value ($t_{3,cr}$), the solutions to equations (S4) become complex and the band degeneracies move outside the Brillouin zone, as illustrated in Fig. 2e, f. Although, the bulk, now, lacks any trace of the Dirac points, their transition to the complex plane is not sufficient by itself to also eliminate the presence of the associated edge states, which, as a consequence, now extend throughout the entire Brillouin zone (Fig. 2g). In general, the position of a Dirac pair can be adapted continuously over any $\mathbf{k}$-space curve (see Supplementary Fig. 1b) but here, for simplicity, we parametrically trailed the lines of Fig. 2f by perturbing a single chain, until the (0, $\pm\pi/2$) points are reached and the cones are merged, annihilated, and carried over to the complex plane. As such, we are able to construct states that start and end between any points in $k_y$.

By introducing the secondary chains (Fig. 2h; red waveguide sites), associated with the $t_s$ hopping term, we cause the Dirac cones to tilt (Fig. 2i). For the properly terminated lattice, we display in Fig. 2j, k the dispersion curves corresponding to different values of $\Delta n_s$. The group velocity ($\partial k_z/\partial k_y$) can be evidently associated with the tilt of the $k_z$-symmetry plane at a Dirac degeneracy point (red line in Fig. 2i); and can be, therefore, related to the magnitude of $\Delta n_s$ ($t_s$). More specifically, above a critical value ($t_{s,cr}$), the system transitions to the type-II phase and the intersection with the Fermi plane becomes conical (see also Supplementary Fig. 1g, h). For this case (Fig. 2j), the type-II edge states are noticeably curved, as the peripheral modes obtain a high transverse group velocity. By increasing $\Delta n_s$ (and correspondingly $t_s$) even further, we can proportionately increase the tilt of the Dirac cones and, as an outcome, the transport speed of the edge states (Fig. 2k). Hence, the transport speed is explicitly associated with an aspect of the Dirac model; an important implication that allows its precise control in the current system.

A combination of the hitherto examined perturbations, i.e., a concurrent variation of $\Delta n_3(t_3)$ and $\Delta n_s(t_s)$, leads to an extended class of edge states that emerge from complex pairs of type-II cones. This can be accomplished for a properly perturbed case in which $t_3 > t_{3,cr}$ and $t_s > t_{s,cr}$. The new tilted states will occupy the entire Brillouin zone, as shown in Fig. 2l for $\Delta n_3$, $\Delta n_s = 1.1\Delta n_0$, retaining the same level of high parametric control. In essence, by progressively disturbing the Lorentz invariance and rotational symmetry independently (first and second term in Eq. (1)), we precisely tailor the edge state characteristics and generate modes with the desired dynamic properties. From beyond a purely physics standpoint, explicitly controlling the group velocity and occupation of states in $\mathbf{k}$-space can be deemed rather compelling for many solid-state or photonic applications that can exploit this class of excitations.

**Observation of dynamic phenomena in the photonic analog.** To experimentally explore the dynamic transport properties of these systems, we employed the femtosecond laser direct writing

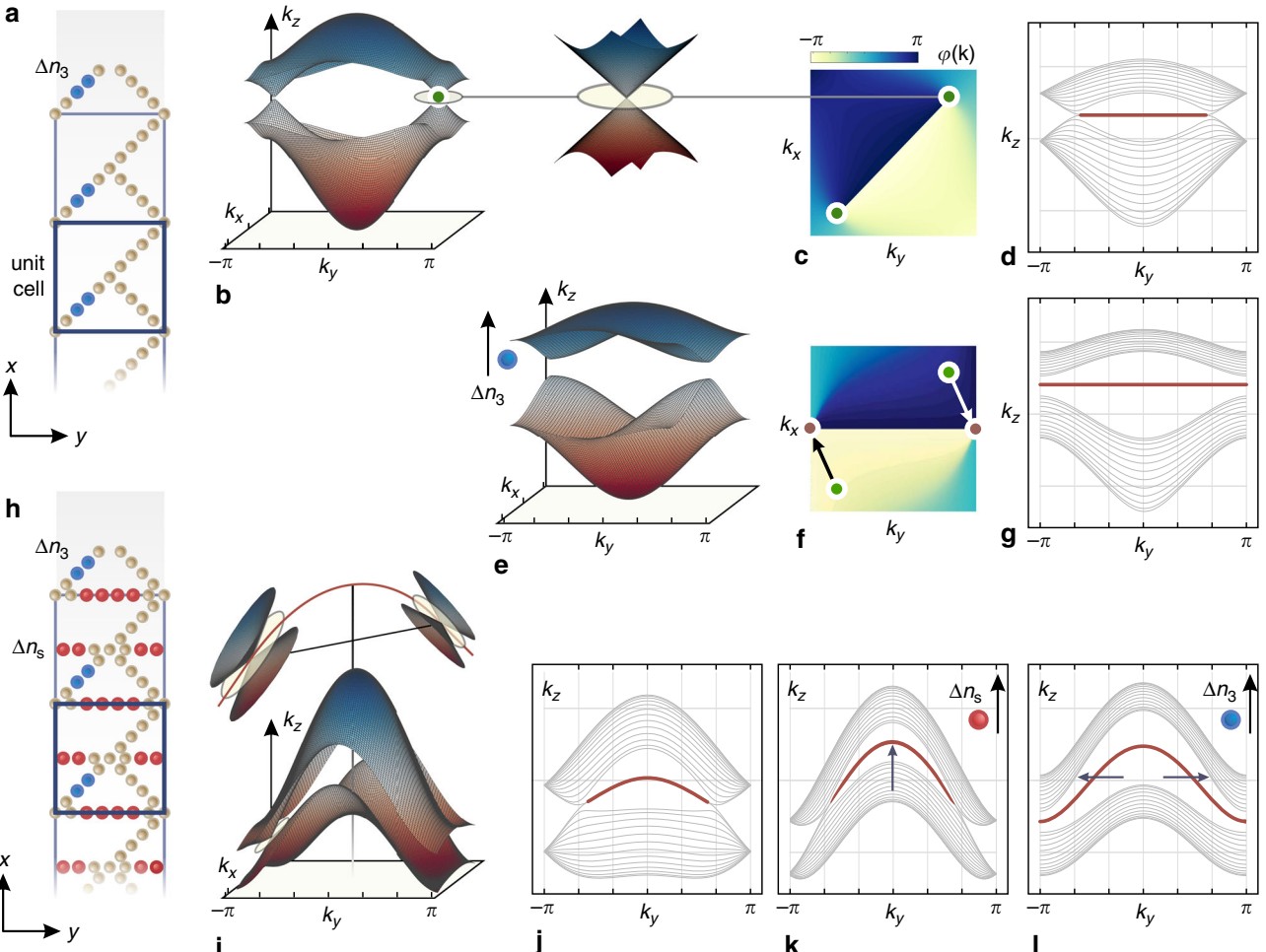

**Fig. 2 Ribbon lattice and band structures of infinite and terminated systems. a** The properly terminated type-I square lattice. The highlighted (blue) sites indicate the waveguides that regulate the hoping term $t_3$. They are associated with a variable index difference $\Delta n_3$. **b** The bulk band structure corresponding to an infinite lattice (both along the x- and y-direction), with a unit cell as shown in **a**, when $\Delta n_3 = \Delta n_O$. **c** The Zak phase term $\phi(\mathbf{k})$ indicating that the edge states should appear along the discontinuous $(-\pi \rightarrow \pi)$ line. **d** Band diagram corresponding to the ribbon lattice of **a** ($\Delta n_3 = \Delta n_O$) with the edge states given in red. **e**–**g** Bulk band structure, Zak phase term $\phi(\mathbf{k})$, and ribbon band diagram corresponding to **a**, yet for $\Delta n_3 = 1.1\Delta n_O$. This higher index value increases the $t_3$ hopping term, resulting into the relocation of the Dirac cones to the red spots of **f**. As a result, the edge states extend now throughout the Brillouin zone, as shown in **g**. **h** The type-II square lattice where secondary s-chains have been introduced. The red and blue sites indicate the waveguides that regulate $t_s$ and $t_3$ (associated with a variable index difference $\Delta n_s$ and $\Delta n_3$, respectively). **i**, **j** Bulk and ribbon band structures when all waveguides are identical ($\Delta n_3 = \Delta n_s = \Delta n_O$). The curvature of the edge states in **j** is associated with the red symmetry line of **i**. Ribbon band diagram for ($\Delta n_3 = \Delta n_O$, $\Delta n_s = 1.1\Delta n_O$) and ($\Delta n_3 = \Delta n_s = 1.1\Delta n_O$). By perturbing the red and blue highlighted waveguides sequentially, we attain full control over the curvature (in **k**) and then extend (in **l**) of the edge states.

technique to inscribe discrete photonic lattices of evanescently coupled waveguides[43] arranged in the respective geometries. The samples were characterized by exciting specific lattice sites at different wavelengths, where, light from a fiber-coupled supercontinuum source (NKT SuperK Extreme) was spectrally selected to a bandwidth of ~2 nm (see also Supplementary Note 5). By varying the excitation wavelength, we were able to change the effective coupling strength between the waveguides, and therefore the rate of the transverse evolution in the lattice. Since the coupling between waveguides arises from the overlap of evanescent fields, it increases strongly with the wavelength. In contrast, the effective refractive index of the individual sites stems from the overlap of the central part of the mode with the waveguide core, and is therefore affected to a much lesser extent. In this vein, we were able to investigate different dynamics regimes while keeping the excitation conditions fixed.

We, first, study the edge dynamics in the basic three-chain model (with no secondary chains attached), with the respective

fabricated prototype presented in Fig. 3a. In this system, we implemented two distinct cases, one corresponding to the band diagram of Fig. 2d in which all 14 waveguides of the unit cell are identical ($\Delta n_O$), and one corresponding to the band diagram of Fig. 2g, in which the perturbed (blue) waveguides are written with a 10% larger refractive index contrast difference ($\Delta n_3 = 1.1\Delta n_O$). In this latter case, the edge mode's dispersion curve extends throughout the entire Brillouin zone, indicating that an edge state can emerge for any wavenumber in $\mathbf{k}$-space. Hence, even narrow excitations that comprise a wide range of $k_y$ values across the first Brillouin zone, will populate this state with high specificity and, under ideal conditions, none of the supported bulk modes. Figure 3c, d illustrates the observed intensity distributions at the end facets of the respective lattices. As indicated in Fig. 3a, the single-site input spans less than one fifth of the unit cell, clearly much wider than $2\pi$ in the normalized $\mathbf{k}$-space. For both cases, we expect that any emergent edge state will remain static due to the vanishing dispersion of the respective mode. Herein, as

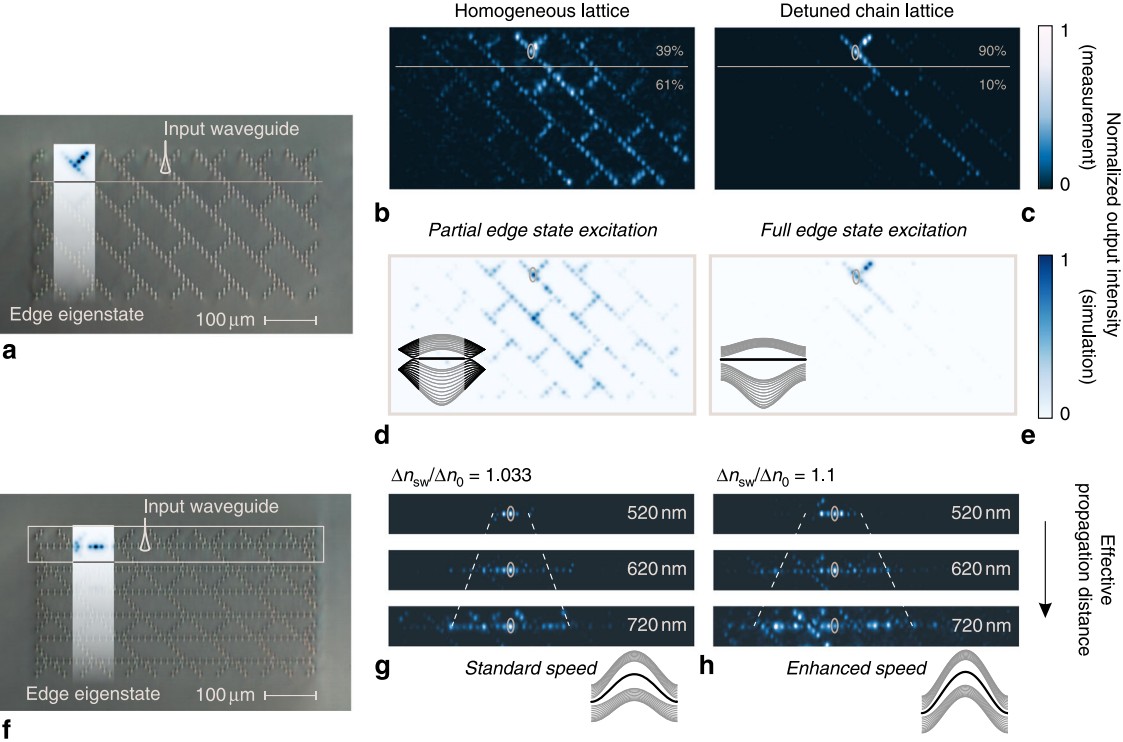

**Fig. 3 Physical models and experimental results.** **a** Micrograph of the three-chain lattices. The numerically calculated edge eigenstate is portrayed as overlay. **b** Observed distribution of light power at the end facet of the homogenous lattice. A considerable amount of power is leaked into the bulk. **c** Light power distribution for the lattice with a detuned chain. The edge mode occupies all **k** values and the edge state is excited almost exclusively. **d**, **e** Simulation results, obtained via the beam propagation method (BPM), for the two experimental lattices which demonstrate an analogous response. **f** Micrograph of the type-II lattice and the numerically calculated edge eigenstate. **g**, **h** Light power distribution at the edge for different wavelengths and different amounts of detuning ($\Delta n_{sw}/\Delta n_0$) for the secondary chains. The different probe wavelengths, effectively change the propagation distance within the physical sample. By adjusting $\Delta n_{sw}$, we modify the curvature of the edge-propagating modes and achieve higher group velocities that result in the observed spread of the pulse (for the respective simulation results see Supplementary Fig. 4).

predicted, the lattice with the enhanced chains (second case) displays a clear formation of an edge state with minimal leakage into the bulk. The leaked fraction of power in the experiment amounts to less than 10% and is attributed to the mismatch between the edge state eigenfunction and the single-waveguide input (see Supplementary Fig. 5) and possibly, to a lesser extent, to the structural imperfections of the lattice. In contrast, the leakage in the lattice without these enhanced chains exceeds 60%, as can be expected given that the edge state here only exists across two thirds of the Brillouin zone ($k < |2\pi/3|$). Notably, while the increased refractive index boosts the coupling strength between the elements and one might expect a corresponding enhancement of the leak effect of Fig. 3b, the tight-binding dynamics, actually, enable a total confinement of light at the edge, as observed in Fig. 3c. This characteristic is of vital importance in the physics of edge state dynamics and, as evident here, can be systematically implemented with only minor modifications of the underlying homogeneous lattice.

The second experiment involves two separate lattices, designed to exhibit type-II Dirac cones that reside in the complex space of **k**. This means that the associated bulk band diagrams will not exhibit Dirac degeneracies but their presence in the complex plane will lead to the emerge of tilted states at the edge. A micrograph of the lattice geometry is depicted in Fig. 3f, replicating the example of Fig. 2h. Here, the addition of the secondary chains grants access to the Lorentz-breaking term of the Dirac Hamiltonian, which serves to control the curvature of the edge modes in the **k**-space. By increasing the refractive index of the waveguides in these chains we direct the transport speed of

the edge modes. We conduct experiments in two lattices with two different refractive index contrasts. In Fig. 3g, h, the introduced curvature in the momentum space leads to the initially confined wave packet being transported symmetrically along the edge. The rate of transport corresponds exclusively to the magnitude of detuning ($\Delta n_{sw}/\Delta n_0$), applied in the secondary chains, and varies proportionally to the number of detuned waveguides, here the middle four. In general, the concurrent modification of a larger group of waveguides leads to a faster response, hence, a better fine-tuning of the mode's transport speed can be achieved by adjusting only the central waveguide pair of the chain.

In addition to the observed cases, the chained lattice may exhibit a number of secondary edge states that have not been discussed yet. For the properly terminated lattice these states emerge from the Dirac cones of lower energy bands and relate directly to the topological characteristics of these cones. They, thus, conform to the same conjectures made for the dominant modes. Nevertheless, as they may interfere with the intended observables, we explore the possibility of isolating them. In the preceding experiment, the light energy was injected directly into the fourth waveguide site of the secondary chain, maximizing the Hermitian inner product with the eigenfunction of the dominant edge state (first modal group of Fig. 4a). Using the same setup, a different site is excited (marked in a circle) so as to more efficiently couple to the secondary edge state (second modal group of Fig. 4a). Evidently, both edge states can be excited independently with no leakage into the bulk. It is noteworthy that for non-proper terminations, the secondary modal groups can manifest simultaneously bearded-like, zigzag-like as well as a

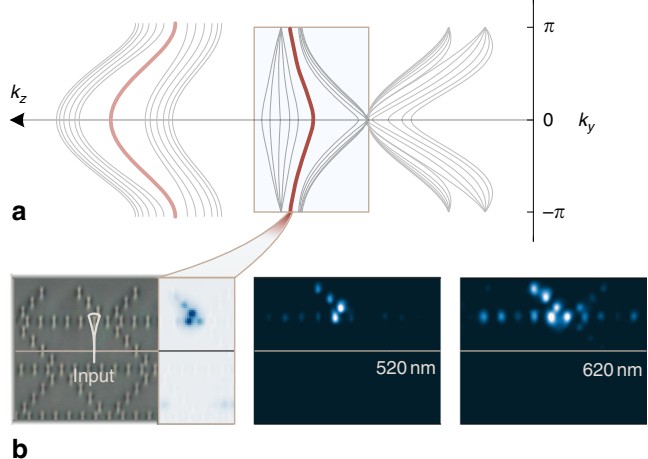

**Fig. 4 Secondary modes. a** Ribbon band diagram of the type-II lattice. The first modal group is associated with the main results of Figs. 2 and 3. Lower energy bands exhibit additional pairs of edge states. **b** By probing a different site, we excite the secondary edge eigenstate shown as overlay on the micrograph panel. The experimental results are captured for two different wavelengths of the laser probe.

number of unconventional edge states (see Supplementary Fig. 3), displaying a rich set of excitations.

## Discussion

In conclusion, we have experimentally observed the excitation and transport of an extended class of type-II edge states in a two-dimensional setting. The proposed tight-binding model allows a controlled variation in its effective lattice intercouplings and provides full access to the parameter space associated with the quasi-relativistic model. The parametrization avoids any distortions to the form or geometry of the unit cell, which, also, preserves the overall dimensions of the lattice itself. It is thereby superior to alternative means of variation, such as the application of strain. Of particular interest, as well, will be the exploitation of nonlinear effects in the photonic analog, so as to manipulate the bonding strength of the chain elements (e.g., by switching the effective refractive index though localized spatial solitons[44]), altering dynamically the state of the Dirac model and its transition between the type-I and type-II phases. Finally, the intrinsic simplicity of the featured chain model could enable different approaches that may lead to the non-trivial topological transition of the type-II cones in particular modified configurations[45].

## Methods

**Experimental configuration**. The photonic lattices are fabricated by focusing ultrashort laser pulses from a Ti:sapphire regenerative amplifier system (Coherent Mira/RegA, wavelength 800 nm, repetition rate 100 kHz, pulse length 130 fs) into the volume of a fused silica sample (Corning 7980, dimensions 1 mm × 20 mm × 100 mm, background refractive index $n_0 = 1.457$ at 633 nm), thereby inducing permanent refractive index changes along arbitrary three-dimensional trajectories as defined by the motion of a precision translation system (Aerotech ALS130, inscription speed 100 mm/min). Owing to the focussing conditions, these waveguides exhibit slightly elliptical mode fields with a typical effective refractive contrast index of $\Delta n_0 = 5 \times 10^{-4}$. The selective positive (negative) detuning of connecting sites was achieved by an appropriate decrease (increase) of the inscription speed.

**Polychromatic probing of the lattices**. The laser-written waveguide lattices were probed by exciting specific lattice sites at different wavelengths. To this end, light from a fiber-coupled supercontinuum source (NKT SuperK Extreme) was spectrally selected to a bandwidth of ~2 nm in a seamlessly tunable fashion (monochromator NKT LLTF). By adjusting the wavelength of the excitation, the effective coupling strength between the waveguides can be increased (longer wavelengths) or decreased (shorter wavelengths) at will, yielding a corresponding acceleration or

deceleration of transverse evolution in the lattice. Notably, the effective refractive index of the individual sites remains largely unaffected, as it mainly stems from the overlap of the central part of the mode with the waveguide core. In this vein, we were able to investigate different dynamics regimes while keeping the excitation conditions fixed.

## Data availability

The experimental data that support the findings of this study are available from M.H. upon reasonable request.

## Code availability

The MATLAB® codes corresponding to the BPM and band structure algorithms are available from G.G.P. upon reasonable request.

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

## Acknowledgements

A portion of this work was supported by the National Science Foundation (NSF: DMR-1420620), the Defense Advanced Research Projects Agency (DARPA: HR00111820042, HR00111820038), the Qatar National Research Fund (QNRF: NPRP9-020-1-006), and the Office of Naval Research (ONR: N00014-18-1-2347). N.S.N. acknowledges the support of the Alexander S. Onassis Public Benefit Foundation and the Foundation for Education and European Culture. Moreover, A.S. thanks the Alfried Krupp von Bohlen und Halbach Foundation for supporting his research. Finally, the authors would like to thank C. Otto for preparing the high-quality fused silica samples employed in all experiments presented here.

## Author contributions

G.G.P. initiated the idea, conceived the experimental setting, and performed the theoretical and numerical work. M.H. and N.S. synthesized the waveguide lattices serving as photonic implementation of the system, and carried out the experiments therein under the supervision of A.S. Moreover, N.S.N. assisted on theoretical and numerical grounds under the supervision of N.V.K. and D.N.C. All authors contributed to writing the manuscript and with critical comments on the project.

## Competing interests

The authors declare no competing interests.
