## [Peer Review File · Nature Communications]

Reviewers' comments:

Reviewer #1 (Remarks to the Author):

The manuscript by G. Pyrialakos et al. have proposed a scheme to control the edge states in 2D Dirac photonic crystal with evanescently coupled waveguides. Compared to previous work [Phys. Rev. Lett. 119, 113901 (2017)] which mainly introduce the bulk properties of the same photonic system, this work uncover the relation of the edge states to the bulk, and provide experiment evidents about the properties of the edge states. The configuration of this system have shown its flexibility controlling the location and anisotropy of the Dirac cones and the shape of the edge states. This work may be considered for publicaiton in Nature Communication after adressing the comments below.

1. The meaning of k_z is not explained in the manuscript and may cause confusion, as k_z is very different from k_x or k_y . In my understanding, k_z is the propagation constant related to the wavelength of light. In Fig. 3 and Fig. 4, the author use wavelength to denote the experiment results. Here the relation between wavelength and certain k_z is not given, making it hard to relate the experiment results to with the edge band structure.
2. In Fig. 3c and 3e, if the excitation frequency is deviated from the exact frequency of edge state but not into the bulk, what would happen?
3. The authors denoted that the power leakage can be attributed to the mismatch between the edge state eigenfunction and the single-waveguide input. However, the power leaked into the bulk shown in Fig. 3c may be caused by mismatching in structure fabrication.

Reviewer #2 (Remarks to the Author):

This paper presents theory and experiments related to photon propagation in arrays of coupled one-dimensional waveguides where the coupling can be carefully tailored to achieve tilted Weyl cones, leading to fascinating new edge states of finite-size lattices. The authors demonstrate the ability to engineer the photonic band structure to replicate key properties of Type-II Weyl systems. The edge modes can have controlled dispersion and extent in the Brillouin zone. The degree of control of the band structure is remarkable, making this a unique study.

The experimental demonstration of the type-II Weyl edge modes is quite compelling. The use of variable wavelength excitation to change the transverse coupling is very elegant. However, Ref. [33] has made experimental observation of Type-II edge modes before. Reference [34] has experimentally examined discrete coupled lattices with a simpler and less facile coupling scheme. This is an elegant extension of theoretical work done by many of the same authors in Ref. 31. I endorse publication of these results in Nature Communications after the following issues have been addressed.

It would be helpful to the readers to give a clearer explanation of the lattice structure. Frankly, I find Fig. 1(a) rather confusing and it needs to be re-imagined. It is a mixture of 2D and 3D features and is supposed to represent a graphyne structure, but it is not clear. What do the thin black lines represent? This lattice structure can be shown much better.

What do the colors of the dots in Fig. 2(a) and (h) represent?

The Methods section should also describe in some detail how the measurements shown in Figs. 3 and 4 are performed.

Reviewer #3 (Remarks to the Author):

In this manuscript, the authors develop an analytical and experimental model for realizing Dirac points whose corresponding conical band structures are 'tilted,' i.e. they are type-II Dirac points. Experimentally, the authors realize this system in an array of evanescently coupled waveguides.

Using this array, the authors claim that type-II Dirac points realize their “own class of edge states,” and that the system as a whole represents a universal platform for the exploration of quantum relativistic effects across a broad range of systems. Overall, I think the technical claims in the manuscript are generally correct, but I do not think the motivations and implications of this work are sufficient to merit publication in Nature Communications. As such, I would recommend to the editor that the authors should resubmit their manuscript to a different journal, such as Scientific Reports or Physical Review A, possibly as a rapid communication.

My troubles with the motivation for this manuscript stem from the following concerns. First, the unit cells of the author’s system contain either 14 or 23 waveguides, depending on whether $t_S = 0$. Given this, and noting the technical constraints on fabricating waveguides deep into the glass substrate, such systems are limited to ~ 3 unit cells in the vertical direction of the array, and only a few effective interaction lengths along the propagation direction between the nodes of the system. (Although the interaction length between any two waveguides in the system is likely rather short, the authors wish to use a two-band model for their system, and given all of the auxiliary waveguides required to tune the coupling constants between these nodes, the effective interaction length between the nodes of their lattice will be quite long.) As such, it is not clear to me that this system really represents a “highly versatile and universal platform for the experimental exploration” of any broad class of systems.

Following on this is a technical concern. In the current version of the manuscript and supplementary info, the authors have not reported enough of the experimental parameters of their system to know what these interaction lengths are. What is the index of the glass, and the index shift of the waveguides? How long is the total waveguide array? These parameters are particularly important because the experimental system the authors are using only has just over 3 unit cells in the vertical direction of the array. Edge states are exponentially localized to their corresponding edge, but 3 unit cells is pretty short. It could be that the bandgap in which the edge states are embedded is sufficiently large such that 3 unit cells is sufficient for edge localization. It could also be that the effective interaction lengths of the system are too long relative to the total propagation distance of the array to see light appear on the opposite edge, in which case this really is not a realization of the edge states of the system. But, at present, insufficient information is reported to make this determination.

My second concern with the motivation for this manuscript is that it is not clear to me how type-II Dirac points realize their “own class of edge states.” What is different about these edge states from what one would find in a typical hexagonal lattice with the correct edge termination? I understand that these states extend across the entire Brillouin zone, rather than between two Dirac points, but this does not seem like a categorically new type of edge state. In addition, these edge states do not offer any of the possible applications that one might find in the chiral edge states of a Chern insulator, as these states are reciprocal.

Finally, I list here a few technical points for the authors to consider.

The wording in the first paragraph of the introduction is a bit strange. Dirac points in a 2D lattice are sources of Berry curvature, but do not (by themselves) lead to non-zero Chern numbers. So the 3rd sentence of the first paragraph probably should be corrected.

Equation (1) has something funny going on. $k \cdot \sigma = k_x \sigma_x + k_y \sigma_y$. Why do you then say that u^D determines the coefficient in front of $k_x \sigma_y$ and $k_y \sigma_x$? I know what Hamiltonian you want to construct in the supplementary material, but something funny is going on between equations S5-S7 and 1 in the main text.

The entire manuscript works with $t_4 = 0$, which means that the tight binding system is nearly a honeycomb system, rather than a square lattice. In fact, when $t_S = 0$, it is a honeycomb lattice. $t_S \neq 0$ then means that you’re including some set of terms which break the chiral symmetry of

the system. As such, you may be doing future readers a favor by pointing these properties out, and doing so may also help streamline some of the discussion.

The supplementary material has enough typos that it currently isn't very useful. For example, Eq. S14 seems to state that $t_{22} = -t_{22}$. What do the authors really mean to say here? In Eq. S7, do you mean $h_{12} = h_{21}^*$? Or is your system actually non-Hermitian?

Reviewer 1

General comments: “The manuscript by G. Pyrialakos et al. has proposed a scheme to control the edge states in 2D Dirac photonic crystal with evanescently coupled waveguides. Compared to previous work [Phys. Rev. Lett. 119, 113901 (2017)] which mainly introduce the bulk properties of the same photonic system, this work uncovers the relation of the edge states to the bulk, and provide experiment evidence about the properties of the edge states. The configuration of this system has shown its flexibility controlling the location and anisotropy of the Dirac cones and the shape of the edge states. This work may be considered for publication in Nature Communication after addressing the comments below.”

Response: We would like to thank the reviewer for his/her constructive comments and for supporting the publication of our manuscript.

Comment 1: “The meaning of k_z is not explained in the manuscript and may cause confusion, as k_z is very different from k_x or k_y . In my understanding, k_z is the propagation constant related to the wavelength of light. In Fig. 3 and Fig. 4, the author use wavelength to denote the experiment results. Here the relation between wavelength and certain k_z is not given, making it hard to relate the experiment results to with the edge band structure.”

Response: We thank the reviewer for pointing out this potential source of confusion. To assist readers on this issue, we have expanded the Supplementary Information by adding a new dedicated section (see Section S.IV). In our system, the light propagates according to the well-known Schrödinger equation with an effective potential term given by $V = 2k_0^2 n_0 \Delta n w_0^2$. Herein, k_0 denotes the free-space propagation constant, corresponding to the wavelength λ_0 in Figs. 3g-h. Parameter k_z differs from k_0 as it refers to the propagation constant of ϵ which is the envelope of the electromagnetic field that constitutes the observable quantity in our system. Its relation to k_x and k_y stems from equation (S24), which implies the imposition of Floquet-Bloch solutions to the eigenvalue problem of equation (S22) (with k_z the eigenvalue). A reader familiar with quantum theory should consider z as the time component in a quantum-mechanical system where k_z assumes the role of energy E (or frequency ω).

An important property, explicitly related to k_z , is the group velocity ($u_g = \partial k_z / \partial k_y$). Given the prior definition of k_z , it represents the potential spread of the pulse envelope at longer observation distances. In our results, we demonstrate how u_g , among other important properties, can be effectively linked to the quasi-relativistic model of equation (1). By tilting the Dirac cones in the bulk, we also tilt the edge band structure (Figs. 2j-l) and experimentally observe non-stationary states (Figs. 3g-h) due to a finite group velocity. It is our belief that this particular interpretation of the band structure is the most crucial. To better reflect these remarks, we now end the caption of Fig. 3. with a new sentence and also start the second paragraph of the “edge states of ribbon-like topologies” section with a sentence that points to S.IV.

Comment 2: “In Fig. 3c and 3e, if the excitation frequency is deviated from the exact frequency of edge state but not into the bulk, what would happen?”

Response: Each excitation was carried out with monochromatic light (of one specific optical frequency or wavelength), which was projected onto a specific waveguide of the input facet. The output pattern on the output facet was then recorded before repeating the process for the next frequency. The propagation dynamics depend entirely on the overlap of the input spatial distribution with the bulk- and edge-eigenmodes, respectively. Here, the excitation waveguide was chosen in order to optimize the overlap with the edge eigenmode, even though it is the nature of point-like wave packets to carry contributions from across the entire band structure. In other words, the experimental setting is not currently suited to separately vary the excitation frequency for the bulk- and edge-state contributions; this would require two independent laser sources, acting at the scale of μm , to excite two separate waveguides. Nonetheless, in theory, there is no constraint that prohibits the emergence and propagation of edge and bulk modes simultaneously in the lattice. If one excited the bulk at a different frequency/wavelength in Fig. 3c, then the proper mode would emerge and exhibit bulk dynamics that would not interfere with the light distribution trapped at the edge.

Comment 3: “The authors denoted that the power leakage can be attributed to the mismatch between the edge state eigenfunction and the single-waveguide input. However, the power leaked into the bulk shown in Fig. 3c may be caused by mismatching in structure fabrication.”

Response: The reviewer is correct in identifying fabrication imperfections as a potential source of power leakage that would not be captured by simulations of the ideal model system. Two types of such imperfections may occur: (i) deviations of the transverse waveguide coordinates from the desired positions and (ii) undulations of the waveguide trajectories as well as variations of the effective refractive indices along the direction of propagation. The former would result in a band structure that slightly differs from the calculated one, whereas the latter would act as external perturbation that may dynamically transfer intensity between the individual modes of the system. In practice, the geometric deviations of the fabricated waveguide lattice are negligible, given the sub-50 nm accuracy of the translation stages used during inscription, which corresponds to perturbation amplitudes more than two orders of magnitude below the characteristic waveguide spacing employed in our system ($\approx 10 \mu\text{m}$). Moreover, given the comparably slow propagation dynamics, and the rapid variations of the lattice are averaged out over the characteristic coupling length of the lattice (detailed by the inverse nearest-neighbor coupling coefficient, on the order of 0.5 cm for the system at hand). Finally, random refractive index modulations caused by fluctuations of the inscription power tend to occur on the scale of microns as well, in line with the repetition rate of 100 kHz and sample translation speeds on the order of 100 mm/min. Modulations on this scale, likewise, tend to be averaged out by the slow wave packet propagation, and mainly result in slightly increased scattering losses (i.e. coupling to the continuum modes of the bulk host material, and not the bulk modes of the waveguide lattice).

Given our long-standing experience with laser-written waveguide arrangements, another source of power leakage is more likely: An imperfect matching of the focal spot of the injection objective with the mode diameter of the individual waveguides. For isolated guides, this merely reduces the overall injection efficiency, whereas for closely spaced waveguides such as in the vicinity of the vertices of the graphyne lattice, neighboring guides may receive a non-zero fraction of the intensity at potentially random phases, thereby slightly reducing the overlap with the targeted edge mode compared to an ideal single-site excitation. Given the limited choice in available numerical apertures of the excitation objectives, such mode mismatch depends considerably on the excitation wavelength.

However, the majority of power leakage in the bulk can be attributed to the mismatch between the edge mode and the single-waveguide excitation. As illustration, we here provide additional simulation results in Figure R1. The edge eigenmode is shown on the left. We indicate the “most important” waveguides as the three sites that contain the majority of light. Numerical simulations for a propagation distance equivalent to the experiments clearly reproduce the observed amount of bulk leakage for a single site excitation. As shown in the remaining panels, double- and triple- waveguide excitations would progressively yield less leakage. To prevent bulk leakage altogether, one would have to perfectly shape the excitation pattern to match the intensity distribution of the edge mode across an entire unit cell of the lattice. While synthesizing such a complex excitation pattern is in principle possible with a spatial light modulator, doing so across a wide range of wavelengths, while ensuring identical injection efficiencies and strictly maintaining the constant phase between all involved waveguide modes, would pose extreme demands on the resolution of the SLM as well as the fine-tuning of all involved components. Fortunately, the edge state dynamics associated with the type-II phase of the Dirac lattice can be clearly observed without the need to resort to intricate adaptive beam shaping procedures.

In the revised version of our manuscript, we have added a remark about possible leakage due to structural imperfections of the lattice to the fourth paragraph of the “observation of dynamic phenomena in the photonic analog” section, and also included the figure below in the Supplementary Information as Fig. S5.

Fig. R1: Dependence of bulk leakage on the excitation pattern (numerical simulations). Whereas distribution of the edge state (a) yields only a limited overlap with a single-site excitation (b) the fraction of light radiated into the bulk decreases if (c) two or even (d) three waveguides are excited in phase.

Reviewer 2

General comments: “This paper presents theory and experiments related to photon propagation in arrays of coupled one-dimensional waveguides where the coupling can be carefully tailored to achieve tilted Weyl cones, leading to fascinating new edge states of finite-size lattices. The authors demonstrate the ability to engineer the photonic band structure to replicate key properties of Type-II Weyl systems. The edge modes can have controlled dispersion and extent in the Brillouin zone. The degree of control of the band structure is remarkable, making this a unique study. The experimental demonstration of the type-II Weyl edge modes is quite compelling. The use of variable wavelength excitation to change the transverse coupling is very elegant. However, Ref. [33] has made experimental observation of Type-II edge modes before. Reference [34] has experimentally examined discrete coupled lattices with a simpler and less facile coupling scheme. This is an elegant extension of theoretical work done by many of the same authors in Ref. 31. I endorse publication of these results in Nature Communications after the following issues have been addressed.”

Response: We would like to thank the reviewer for his/her kind remarks and for his endorsement of our manuscript.

Comment 1: “It would be helpful to the readers to give a clearer explanation of the lattice structure. Frankly, I find Fig. 1(a) rather confusing and it needs to be re-imagined. It is a mixture of 2D and 3D features and is supposed to represent a graphyne structure, but it is not clear. What do the thin black lines represent? This lattice structure can be shown much better.”

Response: We thank the reviewer for pointing out the potential for improvement in the presentation of the lattice, and thus we have restructured the figure accordingly. Now, the new version more clearly illustrates how the unit cell, with its boundaries indicated by horizontal and vertical lines, is embedded within the surrounding lattice. The caption has been also improved.

Comment 2: “What do the colors of the dots in Fig. 2(a) and (h) represent?”

Response: The different colors of the lattice sites serve to indicate the presence (or absence) of detuning. Specifically, the undetuned lattice sites are shown in beige, whereas sites with modified refractive index are shown in blue and red color, for the third main and secondary chains, respectively. The corresponding values of Δn_3 and Δn_s define the magnitudes of hopping terms t_3 and t_s , so, in essence, by detuning the corresponding waveguides, we regulate the edge mode properties, as shown in the band diagrams of Fig. 2. To more effectively convey this meaning, we have accordingly amended the respective figure and caption with three new sentences. In Fig. 2 we now explicitly state the Δn parameters in subfigures e,k and l.

Comment 3: The Methods section should also describe in some detail how the measurements shown in Figs. 3 and 4 are performed.

Response: We agree with the reviewer that more details on the experiments may be of interest for the readership, and have therefore expanded the methods section with pertinent parameters and additional information on the measurement procedures, as well as introducing a new dedicated section (S.V) to the Supplementary Information.

Reviewer 3

General comments: “In this manuscript, the authors develop an analytical and experimental model for realizing Dirac points whose corresponding conical band structures are ‘tilted,’ i.e. they are type-II Dirac points. Experimentally, the authors realize this system in an array of evanescently coupled waveguides. Using this array, the authors claim that type-II Dirac points realize their “own class of edge states,” and that the system as a whole represents a universal platform for the exploration of quantum relativistic effects across a broad range of systems. Overall, I think the technical claims in the manuscript are generally correct, but I do not think the motivations and implications of this work are sufficient to merit publication in *Nature Communications*. As such, I would recommend to the editor that the authors should resubmit their manuscript to a different journal, such as *Scientific Reports* or *Physical Review A*, possibly as a rapid communication.”

Response: We would like to thank the reviewer for devoting the time to evaluate our manuscript and for providing extensive feedback. In the following, we address each of the points raised in his/her review and detail the improvements to the manuscript inspired by them.

Comment 1: “My troubles with the motivation for this manuscript stem from the following concerns. First, the unit cells of the author’s system contain either 14 or 23 waveguides, depending on whether $t_S = 0$. Given this, and noting the technical constraints on fabricating waveguides deep into the glass substrate, such systems are limited to ~ 3 unit cells in the vertical direction of the array, and only a few effective interaction lengths along the propagation direction between the nodes of the system. (Although the interaction length between any two waveguides in the system is likely rather short, the authors wish to use a two-band model for their system, and given all of the auxiliary waveguides required to tune the coupling constants between these nodes, the effective interaction length between the nodes of their lattice will be quite long.) As such, it is not clear to me that this system really represents a “highly versatile and universal platform for the experimental exploration” of any broad class of systems.”

Response: The reviewer correctly indicates the high number of waveguides in the two types of complex unit cells, and we wholeheartedly agree that the femtosecond laser direct writing technique employed for sample fabrication imposes certain limits on the overall system size as well as the rate of the transverse evolution. However, we have to respectfully disagree with the conclusions drawn on this basis. Even the comparably small lattice size of the 3×6 unit cells is sufficient to faithfully capture the characteristic behavior of the systems under consideration, namely the emergence of strongly confined edge states within the predicted band gap, as well as the transverse propagation dynamics within the channels defined by them. The reasons for this are two-fold.

For one, as long as the propagating wave packet does not reach the boundaries of the system, it remains entirely agnostic as to its overall size, and therefore exhibits evolution dynamics that are indistinguishable from the infinite configuration. On a more general note, this fact is quite fortuitous, as it constitutes the basis for any and all experiments in such discrete systems that – in their real-world implementation – have to be limited to a certain finite size. On the other hand, the exponential localization of the edge states allows them to be efficiently populated by single-site excitations, and, for the propagation distances at hand, effectively shields them from interactions with the edge states residing at the opposite edge of the lattice, even if they are only separated by a few unit cells. The size and geometry of our experimental settings were chosen after an extensive evaluation process that, relying heavily on numerical simulations, ensure the validity of these assumptions (see also below for some indicative examples).

In line with these considerations, we included the statement of a “highly versatile and universal platform for the experimental exploration” not with respect to the fabrication technique, but relating to the discrete graphyne-like lattice itself. Being comprised of individual single-moded waveguides, the photonic model enables a straight-forward design protocol for the generation of Dirac systems exhibiting a type-II phase with its desirable properties. Being related only to the discrete underlying lattice arrangement and its tight-binding formalism, the same methods can be applied to other discrete systems in photonics (such as coupled microring resonators, optically induced photorefractive lattices or multicore fibers) as well as other fields of physics (lattices of ultracold atoms, microwave cavities, electrical circuits or molecular crystals, to name a few). As such, the photonic analog constitutes merely a proof of certification and applicability for the proposed methodology, rather than an exclusive platform (i.e. no hidden variables or synthetic dimensions are being used specific to photonic lattices). We thank the reviewer for helping us recognize that the phrasing in the previous version of the manuscript perhaps did not convey the full scope of this. To resolve it, we partially rephrased the last sentences of paragraph 3 in the introductory section as well as a small part in the conclusions paragraph.

Comment 2: “Following on this is a technical concern. In the current version of the manuscript and supplementary info, the authors have not reported enough of the experimental parameters of their system to know what these interaction lengths are. What is the index of the glass, and the index shift of the waveguides? How long is the total waveguide array? These parameters are particularly important because the experimental system the authors are using only has just over 3 unit cells in the vertical direction of the array. Edge states are exponentially localized to their corresponding edge, but 3 unit cells is pretty short. It could be that the bandgap in which the edge states are embedded is sufficiently large such that 3 unit cells is sufficient for edge localization. It could also be that the effective interaction lengths of the system are too long relative to the total propagation distance of the array to see light appear on the opposite edge, in which case this really is not a realization of the edge states of the system. But, at present, insufficient information is reported to make this determination.”

Response: We have updated the Supplementary Information with the experimental parameters: The fused silica serving as host material for our waveguides has a bulk refractive index of $n_0 = 1.457$ at 633nm. Estimated from numerically inverting the mode profile, the waveguides exhibit a refractive index contrast on the order of $\Delta n_0 = 5 \cdot 10^{-4}$, relative to which the desired detunings were implemented by varying the inscription speed. At 633nm, the elongated, approximately supergaussian index profiles result in slightly elliptical mode fields measure approximately $6 \mu\text{m} \times 8 \mu\text{m}$. The sample length is 100mm, and the inscribed lattices are comprised of three by six unit cells, respectively. In addition to the first paragraph of the “Observation of dynamic phenomena in the photonic analog” section of the manuscript, and its methods section, where this information was initially displayed, it has now also been included in the Supplementary Information S.V for the convenience of the reader.

Regarding the question of whether a genuine edge state is obtained, we would like to point out that by virtue of their exponential localization, the edge states in any finite-width ribbon will always hybridize due to the non-zero overlap of their evanescent tails extending across the bulk. As such, the distinction of confined edge states and interacting channels in a real physical system becomes moot, or, at the very least, is always a question which length scale is predominant – the one that drives the dynamics within the edge state channel, or the one that mediates the interaction between edge states residing on opposite sides of the system.

From a theoretical standpoint, the fact that a “local” band structure [Hadad *et al.*, *Phys. Rev. B* **93**, 155112 (2016)] can be defined for the region that contains a wave packet means that even a small number of unit cells is capable of sustaining the general features predicted by the ideal band structure which, per definition, can only be calculated for (semi-)infinite systems. As stated above: For as long as the propagating wave packet does not reach the boundaries of the system, its dynamics are indistinguishable from the ideal infinite configuration. It readily follows that the dynamics at the edge itself remain unperturbed by the influence of the opposite edge until the parts of the wave packet that did encounter the opposite side have propagated back across the ribbon once more. Whereas the former condition may indeed begin to be violated at infrared wavelengths, the latter condition is definitely fulfilled for the sample lengths at hand, meaning that the observed behavior faithfully represents the theoretically predicted dynamics at the excited edge.

To substantiate our claims, we provide a comparative simulation study in the Figs. R2 and R3. We demonstrate the propagation dynamics of light for three different arrangements of the type-II chained lattice. In all cases, we excite both the edge and bulk of the lattice by inducing light on a single waveguide at the appropriate site. The results are as follows:

- If an edge state is supported by the lattice, light injected at the edge remains strongly confined there (Fig. R2). As there is virtually no difference between the edge excitations in the 3×10 and the 8×10 arrangements even at ten times the propagation distance used in our experiments, the total number of three cells in the experimental setting can be regarded as sufficient.
- The simulations depicted in Fig. R3 illustrate the propagation dynamics in the edge channel. Notably, the two lattices comprised of 3×6 and 3×10 unit cells display identical behavior (Figs. R3a,b) until the wave packets reach the left and right boundaries. From that point on, they backscatter (Figs. R3c-e), but nevertheless remain at the edge. These simulations show that for the experimental sample length, the boundaries of the 3×6 lattice have not yet been reached (Fig. R3b), proving the comparison in Figs. 3g,h to be technically sound.

Fig. R2: Numerically simulated output patterns resulting from single-site excitations of extended type-II lattice ribbons. (a) Edge- and (b) bulk excitation in a ribbon comprised of 3×10 unit cells. (c,d) Corresponding results for a ribbon of 8×10 unit cells.

Fig. R3: Numerically simulated propagation in the edge channel for system sizes of 3×6 and 3×10 unit cells, respectively. Shown are the respective edge channels only. (a) Snapshots of the evolution from a single-site excitation after (a) 30%, (b) 100%, (c) 150%, (d) 200% and (e) 1000% of the propagation distance corresponding to our experiments.

Comment 3: “My second concern with the motivation for this manuscript is that it is not clear to me how type-II Dirac points realize their “own class of edge states.” What is different about these edge states from what one would find in a typical hexagonal lattice with the correct edge termination? I understand that these states extend across the entire Brillouin zone, rather than between two Dirac points, but this does not seem like a categorically new type of edge state. In addition, these edge states do not offer any of the possible applications that one might find in the chiral edge states of a Chern insulator, as these states are reciprocal.”

Response: A resemblance between the edge states emerging in a honeycomb lattice and those that emerge in the chained lattice may indeed occur. In fact, both are topologically related to the presence of Dirac pairs in momentum space, and, therefore their attributes are explicitly associated with the form of the effective relativistic model. However, the two systems differ from each other through this common property. Essentially, the chained lattice constitutes a much more complete simulator of the quasi-relativistic model in 2D, arguably more so than any other tight-binding system (see also the response to Comment 6). Herein, we can define the parameter space of Hamiltonian (1) as the space spanned by the eight variables of the u^D matrix and u^T , \mathbf{k}_D vectors. In Section 1 of the Supplementary Information, we systematically prove that the chain methodology allows us to construct any form of equation (1), and particularly define Dirac cones with any combination of anisotropy or tilt, at any point in momentum space. This trait is effectively carried over to terminated systems and their emergent edge states.

Ultimately, the edge states of honeycomb lattices emerge as a subclass of the broad group of states that we observe here. Yet, the conventional honeycomb lattice cannot generate type-II cones or support their associated tilted states (as the ones we demonstrate in Figs. 2j,k,l and Figs. 3g,h.). While states with finite group velocity are not uncommon in two-dimensional settings, the fact that this property is now associated with an aspect of the Dirac model (the magnitude of the Lorentz violating term) and can thus be readily and directly controlled, is a significant, in our opinion, implication by itself. These states will still always extend between two Dirac

points, as occurs in Fig. 2l, where we expanded the type-II cones to the complex space of solutions defined by equation (S4) and therefore moved them outside the first Brillouin zone. This is done independently by varying the main chains, so we can achieve tilted states that may start and end between arbitrary pairs of points in k_y . Such a trait allows us to excite the edge by injecting light into a single waveguide, while an unperturbed honeycomb lattice would require a broad pulse. Nonetheless, this trait is not the key point of interest, but just an auxiliary property, exploited to enhance the fully controllable tilted states of Fig. 3g,h. In this figure, we now display the respective band diagrams to make this association clearer. In addition, we thoroughly revised most sentences in the three final paragraphs of section “The edge states of ribbon-like topologies” to better convey the previous remarks. Finally, we added a new sentence in the third paragraph of section “Observation of dynamics phenomena in the photonic analog”.

Comment 4: “Finally, I list here a few technical points for the authors to consider. The wording in the first paragraph of the introduction is a bit strange. Dirac points in a 2D lattice are sources of Berry curvature, but do not (by themselves) lead to non-zero Chern numbers. So the 3rd sentence of the first paragraph probably should be corrected.”

Response: The reviewer is correct in his remark that a Dirac cone cannot lead by itself to a non-zero Chern number. The Chern number is a global quantity that may obtain non-trivial values when the Berry phase integral acquires the same sign for both cones in a Dirac pair (for example, due to a broken time-reversal symmetry). In order to avoid any ambiguity, we have rephrased the sentence in question accordingly.

Comment 5: “Equation (1) has something funny going on. $k \cdot \sigma = k_x \sigma_x + k_y \sigma_y$. Why do you then say that u^D determines the coefficient in front of $k_x \sigma_y$ and $k_y \sigma_x$? I know what Hamiltonian you want to construct in the supplementary material, but something funny is going on between equations S5-S7 and 1 in the main text.”

Response: Equation (1) has been enhanced with a pair of brackets to better signify the order of operators. In the revised manuscript it is, now, expressed as

$$H(\mathbf{k}) = u^T \cdot (\mathbf{k} - \mathbf{k}_D) \mathbf{I} + [u^D(\mathbf{k} - \mathbf{k}_D)] \cdot \boldsymbol{\sigma} + m' \sigma_z.$$

The second term should, now, operate in the following fashion

$$\begin{aligned} [u^D(\mathbf{k} - \mathbf{k}_D)] \cdot \boldsymbol{\sigma} &= \begin{bmatrix} u_{xx}^D & u_{xy}^D \\ u_{yx}^D & u_{yy}^D \end{bmatrix} \begin{bmatrix} k_x \\ k_y \end{bmatrix} \cdot \boldsymbol{\sigma} = \begin{bmatrix} u_{xx}^D k_x + u_{xy}^D k_y \\ u_{yx}^D k_x + u_{yy}^D k_y \end{bmatrix} \cdot \begin{bmatrix} \sigma_x \\ \sigma_y \end{bmatrix} \\ &= u_{xx}^D k_x \sigma_x + u_{xy}^D k_y \sigma_x + u_{yx}^D k_x \sigma_y + u_{yy}^D k_y \sigma_y, \end{aligned}$$

where $k_x = k_x - k_x^D$, $k_y = k_y - k_y^D$ is introduced for brevity. This accurately implies the existence of the $u_{xy}^D k_y \sigma_x$ and $u_{yx}^D k_x \sigma_y$ terms, which also appear in equation (S3) (if linearized around \mathbf{k}_D).

Comment 6: “The entire manuscript works with $t_4 = 0$, which means that the tight binding system is nearly a honeycomb system, rather than a square lattice. In fact, when $t_S = 0$, it is a honeycomb lattice. $t_S \neq 0$ then means that you’re including some set of terms which break the chiral symmetry of the system. As such, you may be doing future readers a favor by pointing these properties out, and doing so may also help streamline some of the discussion.”

Response: The reviewer correctly recognizes the absence of the t_4 ancillary branch, yet this does not necessarily restrict the classification to the group of honeycomb lattices. From the perspective of symmetry groups, the lattice is still square and all high symmetry points in the first Brillouin zone are defined accordingly. The tight-binding representation of the lattice is indeed isomorphic to the honeycomb (more profoundly around the Dirac points and in respect to equation (1)), yet only for a very specific set of parameters. The chained lattice actually defines a group isomorphism with every general perturbation of the honeycomb unit cell (e.g. strained graphene). More so, the parameter subspace of equation (1) accessed by the three-chained lattice is substantially larger than the subspace accessed by the group of perturbed honeycomb lattices. The presence of the fourth chain does not fundamentally change the lattice topology or inhibit the emergence of the Dirac pair, on the contrary – by including it, we can unlock even more variations through equations (S6)-(S9). Ultimately, the only resemblance that remains with the honeycomb unit cell is the presence of the three hopping variables

(instead of four) which are now completely disentangled from each other. An indicative consequence of this disentanglement is the ability to relocate the Dirac pair anywhere within the first Brillouin zone (see Fig. S1b), while the group of honeycomb lattices restrains them on the high symmetry lines in momentum space. In general, a symmetry-based approach can lead to the definition and characterization of much richer 2D Dirac systems, beyond the simple honeycomb lattice [Miert *et al.*, *Phys. Rev. B* **93**, 035401 (2016)]. The chained lattice encompasses a large number of features associated with these systems. Concerning the edge states, they are indeed classified as bearded and zig-zag, in line to the graphene terminology, but their mathematical source is more general (see S.II)

The t_s term does not break the chiral symmetry, as this would violate the requirements of equation (1) and therefore introduce a gap (there is none in Fig. 2i). Actually, this term enters under the same sign in the diagonal of our Hamiltonian ($h_{11} = h_{22}$). As it turns out, this is in fact one of the *most fundamental properties* attributed to the chain methodology, enabling virtually any imaginable perturbation of equation (1) and the explicit emergence and variation of the type-II Dirac cones. These remarks are now expressed in the section “Dirac Hamiltonian in a chained lattice model”. The second paragraph has been rephrased almost entirely and supplemented with new sentences.

Comment 7: “The supplementary material has enough typos that it currently isn’t very useful. For example, Eq. S14 seems to state that $t_{22} = -t_{22}$. What do the authors really mean to say here? In Eq. S7, do you mean $h_{12} = h_{21}$? Or is your system actually non-Hermitian?”

Response: We would like to thank the reviewer for identifying a number of typographical errors in the Supplementary Information. Indeed, the system is not Hermitian, so we added the missing “*” on h_{12} . Regarding the t_{22} term, this has been replaced with $t_{22} = -t_{12}$. Considering the reviewer’s general thoughts on the supplementary material we completely restructured Section I. We believe that these modifications will now make our formalism more accessible to the readership, better convey all aspects of our theoretical foundation and also, assist the comprehension of the following sections. More specifically, S.I now starts with an introduction to the 2x2 Hamiltonian and then follows with a description of the chained model. We have also added equation (S14) and the two following paragraphs to enhance the theoretical discussion.

REVIEWERS' COMMENTS:

Reviewer #2 (Remarks to the Author):

I am satisfied with the improvements made by the authors, and I endorse publication of this paper in Nat. Commun. The changes to Fig. 1(a) and the caption are very helpful for the reader. Likewise the changes to the caption of Fig. 2 are also helpful. The additional information in the Methods section and Supp. Mat. about how the experiment is performed is a significant improvement and makes this a complete and reasonably self-contained work.

Reviewer #3 (Remarks to the Author):

I appreciate the authors' responses to my criticism of their manuscript, and overall I think that the changes they have made help tighten the manuscript. Again, I generally think the technical aspects of the manuscript are correct, their underlying mathematical theory seems to be true, and their experimental execution appears to be a faithful representation of that model. However, I still cannot recommend this work for publication in Nature Communications, as I do not think that this work is particularly significant, and I do not think that this manuscript will significantly influence thinking in the field. As such, I again recommend that the authors consider submitting their manuscript to a different journal, such as Scientific Reports or Physical Review A.

The authors are seeking to justify their work by noting that type-II Weyl points have been recently observed in a variety of semimetals, and thus there is an urgent need to understand the properties of type-II Weyl and Dirac points. To this justification for the urgency of this work, I have a few responses:

1) The semimetals the authors have listed all possess type-II Weyl points, and type-II Weyl points have already been observed in waveguide arrays, see Noh et al., Nature Physics 13, 611 (2017).

2) The manuscript only considers type-II Dirac points, and the waveguides in the experimental system are completely straight, there is no chance of creating the necessary 3D crystal structure required to realize Weyl points.

3) It still is not clear to me how type-II Dirac points are significantly different from type-I Dirac points. The authors have attempted to answer this in their reply to my first report, and my understanding of their answer is that the "chained lattice" that they study here is simply a more general system, and so it can realize a broad array of different phenomena, and that I should consider the honeycomb lattice (and its type-I Dirac points) as a special instance of the chained lattice. Fine. But in their more general system, what is new? Sure, they can tilt their Dirac points to make them type-I or type-II. This doesn't seem to dramatically change the physical properties of the system.

Given this, it really does seem to me that this work is just a nice generalization of previously known results, but that this work will not substantially influence future thinking in the field, and as such is more suited to a different publishing venue.